# Intelligent diagnosis and prediction of turbine digital electro-hydraulic control system faults: Design and experimentation

**Ling Zhong**[1], **Qing Li**[2]*

**1** Sichuan Engineering Technical College, Deyang, China, **2** Dongfang Electic Corporation, Deyang, China

* liqing@dongfang.com

**Data Availability Statement:** All relevant data are within the paper and its Supporting Information files.

## Abstract

A physical modeling approach was adopted to build a Digital Electro-Hydraulic Control (DEH) system simulation model and the fault models using the SIMULINK tool. This research combined the advantages of the gray system and neural network to build a multi-parameter gray error neural network fault prediction model for the first time. Furthermore, an embedded platform for intelligent fault diagnosis and prediction was developed using an Application Specific Integrated Circuit chip. The results show that the simulation model of the DEH system has good performance. A jam fault, internal leakage, and a device fault could be accurately identified through the fault diagnosis model. The multi-parameter gray error neural network prediction model improves the accuracy of fault prediction. The embedded platform developed by the Application Specific Integrated Circuit chip solves the problem of transmission limitation and insufficient computing power. It realizes the intelligent diagnosis and prediction of DEH system faults and guarantees the regular operation of the DEH system.

## Introduction

As one of the three major engines of a thermal power plant, a steam turbine is the prime mover with steam as the working fluid and can be called the heart of a thermal power plant. The abnormal shutdown of steam turbines causes a 30% failure of the regulating system [1]. However, failure of the steam turbine regulating system originates mostly in the digital electro-hydraulic (DEH) control system, oil supply system, electro-hydraulic converter, and oil engine. It is crucial to carry out full-cycle control of the steam turbine operation to ensure the quality of the power supply and power production safety [2]. Steam turbines work in a particular environment of high temperature, high pressure, and high speed. During the operation cycle, precise control of the control system and regular maintenance are required [3]. In the industry's early days, steam turbines were controlled utilizing mechanical and hydraulic adjustment systems and electrical hydraulic adjustment [4]. Due to steam turbines' increased capacity and function, the original mechanical–hydraulic and electro-hydraulic control systems are unsustainable [5].

With the rapid development of solid-state electronics technology and new hydraulic technology, the DEH system for steam turbines makes up for the defects of traditional mechanical–hydraulic and electro-hydraulic control systems. It represents the highest level of current

**Funding:** The authors received no specific funding for this work.

**Competing interests:** NO authors have competing interests.

steam turbine control technology. The advantage of the DEH system is that it takes full advantage of the accuracy of the computer. It effectively combines the benefits of the hydraulic system and precisely controls complex equipment by utilizing configuration control software. By controlling the critical parameters of the steam turbine, such as speed and power, the safe production of the steam turbine is guaranteed.

As a control system, the key to the technology is achieving stable operation via computer. Many studies have focused on finding the factors that affect the system's stability through the simulation model, for example, because the model and parameters are not accurate enough in the simulation stability calculation and the problem of how to select the appropriate speed control mode and parameter-setting optimization scheme [6]. Aytac proposed a wind power prediction model based on long short-term memory (LSTM) network and decomposition methods with grey wolf optimizer (GWO) [7]. To accurately simulate the dynamic characteristics of the DEH system, Liao [8] proposed a new feedback control system combined with an artificial neural network, which could effectively identify linear parameters and nonlinear parameters. Using fuzzy neural networks (fuzzy-NNs) to diagnose sensor faults in steam turbine DEH systems was proposed by Mariusz Pawlak et al. [9]. Jin [10] established a steam turbine governor controller for the DEH system, changed the delay time of each component of the DEH system, and simulated the impact of delay failure on the DEH system. In addition, some scholars have perfected the defects of the control system by combining the advantages of other technologies.

Research on fault diagnosis of the steam turbine DEH system has focused on the parameter estimation method of the analytical model and the fuzzy fusion method, but few achievements have been made in fault prediction [11]. The steam turbine DEH system involves many aspects of machinery, electricity, and hydraulics, and its complexity greatly enhances the difficulty of on-site fault diagnosis. However, the existing maintenance and diagnosis methods are relatively backward, and fault prediction is almost completely lacking. Therefore, both an effective fault diagnosis method and intelligent prediction are imperative [12]. The research methods and types of other scholars are not rich, and there are many shortcomings. They have done some work, which involves collecting data on faults that have occurred and analyzing the causes of the faults [13]. But their fault diagnosis methods mostly diagnose vibration faults of steam turbines, and even this diagnostic tool becomes helpless when encountering unknown types of faults [14]. Other scholars' research tools did not take the Mega Data into account, and did not use the collected data to predict the occurrence of failures and prevent them. The first step, we build a DEH system simulation model using the SIMULINK tool. Then, in this paper, by combining data and model. We propose a new method for diagnosing the fault types of DEH systems by extensive data analysis. This can make up for the defects in model analysis, control the component parameter information in the control system in real-time. Finally, we proposed a prediction model of DEH system failure for the first time. Through this model, we can accurately know the probability of failure in the DEH system of the turbine, and know when it will stop working, and maintain the safe operation of steam turbines and ensure the safety of power production.

## DEH system model

### Model construction

The DEH system is essential for the start-up and operation of the steam turbine, and establishing a system simulation model is the cornerstone of the research. Fig 1 shows the steam turbine DEH system's composition, including an oil supply system, an electro-hydraulic converter, a quick unloading valve, a single-side oil motor, and a demodulator [15].

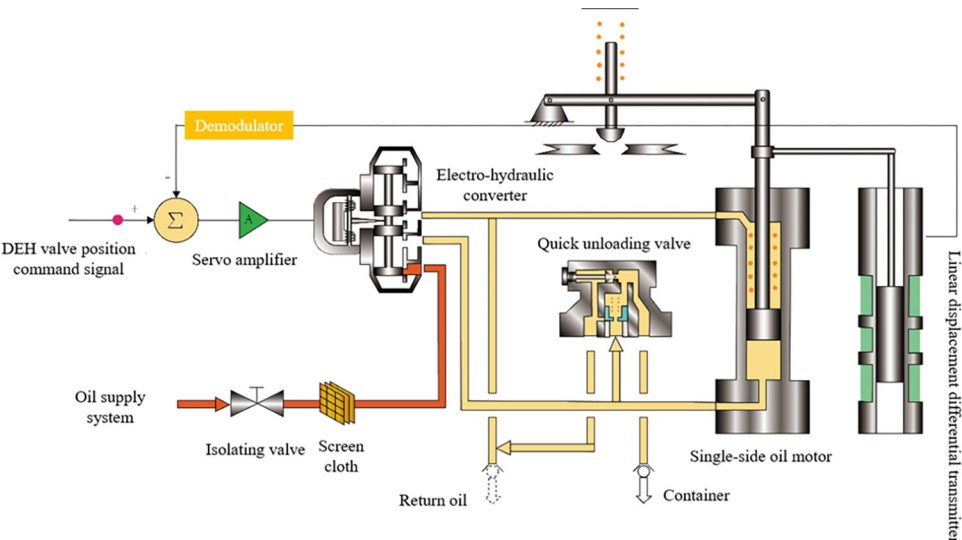

**Fig 1. Schematic diagram of the DEH system structure.**

The steam turbine DEH system simulation mainly includes an equation-oriented algorithm and is physical object oriented [16]. Physical object modeling was chosen for this study. According to the physical operation principle of the research object, the SIMULINK platform was used to build the DEH system simulation model. This makes the model more intuitive and easier to extend and maintain [17]. The construction of the SIMULINK simulation model is shown in Fig 2. Five subsystems were set up according to the fundamental working principle of the steam turbine DEH system. Each subsystem consisted of input, data processing, and output. After the external data were input into the module, the module performed specific processing on the data, calculated the state quantity of the device corresponding to the module, and output the results.

The electro-hydraulic servo valve drives the motor through current, and the motor drives the baffle to rotate, causing the oil pressure at both ends of the spool valve to be different, which further causes the spool valve to move and output a certain amount of oil [18]. Two relationships—the relationship between the current and the displacement of the spool valve and the relationship between the spool valve removal and the output oil volume—are mainly

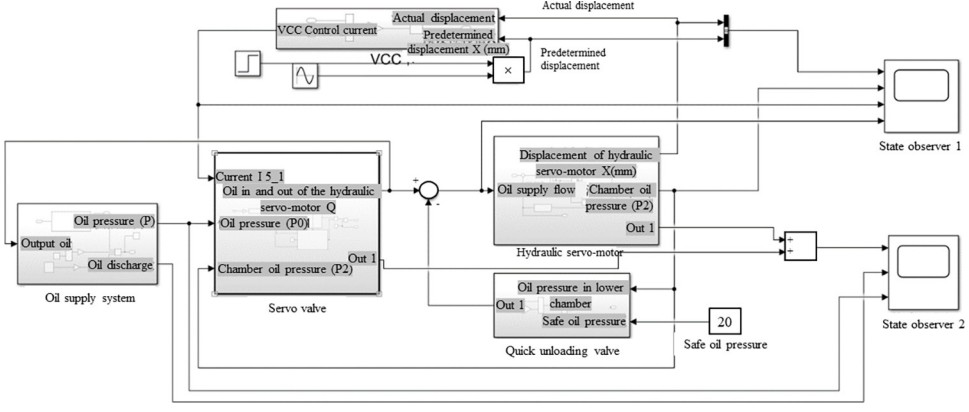

**Fig 2. Construction of the SIMULINK simulation model.**

analyzed when modeling [19]. The mathematical model is as follows:

$$G(\frac{s^2}{\omega^2} + \frac{2\zeta s}{\omega} + 1)\theta + (r + b)k_f X_v + rA_n P_{lp} = k_t I \tag{1}$$

$$X_v = \frac{k}{\frac{s^2}{\omega^2} + \frac{2\delta}{\omega}s + 1}i \tag{2}$$

$$Q = C_d A_0 \sqrt{2/\rho}\sqrt{p_1 - p_2}, \; A_0 = \omega X_v \tag{3}$$

*where θ indicates the deflection angle; G, ζ are the total stiffness and damping coefficient,*
respectively; *ω* is the angular frequency of the oscillating link of the servo valve, *ω = 340rad/
sec; I* is the input current; $X_v$ is the displacement of the spool valve of the servo valve; *δ* is the
damping coefficient of the electro-hydraulic servo valve; *k* is the static gain coefficient of the
electro-hydraulic servo valve; *i* is the coil current; $C_d$ is the orifice flow coefficient; *ρ* is the fuel
resistance density; and *p* is the oil chamber pressure. Considering the actual operation of the
servo valve, the oil inlet and the oil outlet are divided into two kinds of modeling. Among
them, the input is current *i* and the output is oil quantity *Q*.

Oil engine: When modeling, the relationship between the input oil quantity and the dis-
placement of the oil motor and the relationship between the input oil quantity and the oil pres-
sure in the lower chamber of the oil motor are mainly analyzed.

$$Q_3 = A_p + V_p \tag{4}$$

$$m_p v_p = p_f A_p - xv_p - C - kx_p \tag{5}$$

*where* $Q_3$ *is the flow rate of the oil supply;* $A_p$ *is the area of the piston;* $v_p$ *is the velocity of the
piston;* $x_p$ *is the piston displacement; k is the spring elasticity coefficient; C is the spring pre-
load, and* $p_f$ *is the piston pressure.*

Oil supplying system: The oil supply system mainly provides a stable oil pressure. When the
oil supply–demand is high, the output oil pressure fluctuation also increases. The output oil
supply pressure consists of three parts: pressure under normal circumstances, stress caused by
other factors, and the influence on oil pressure caused by the change in oil pump displacement
[20]. The oil supply system is divided into piston pumps and accumulators.

$$Q_L = A_h \frac{dx_p}{dt} + C_{ip}P_c + \frac{V_c}{\beta_e}\frac{dP_c}{dt} \tag{6}$$

$$P_c A_h - P_s A_r = m\frac{d^2 x_p}{dt^2} + B_p \frac{dx_p}{dt} + Kx_p + F_L \tag{7}$$

$$Q = K_f nX_P \tag{8}$$

$$A(p_a - p) = K\frac{V}{A} + \frac{CdV}{Adt} \tag{9}$$

$$A(p_a - p_b) = m\frac{d^2 V}{Adt^2} + \frac{BdV}{Adt} \tag{10}$$

Formulas (6), (7), and (8) can obtain the relationship between the input current and the oil

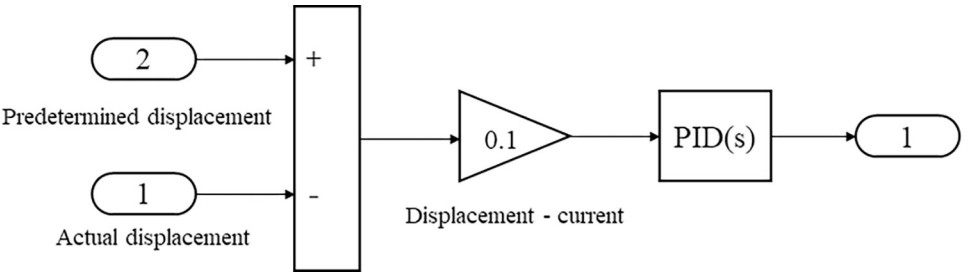

**Fig 3. Volt–current condenser (VCC) servo control loop card model.**

quantity displacement: $Q_L$ is the Control volume; $A_h$ is the piston area; $C_{ip}$ is the internal leakage coefficient of the oil engine; $X_p$ is the piston displacement; $\beta_e$ is the coefficient of volume compressibility; $K_q$ is the flow gain; $K_f$ is the discharge coefficient; and n is the motor speed. $F_L$ is the oil pressure. $B_p$ is the viscous damping coefficient.

Formula (9) is the force balance equation of the air cavity, and Formula (10) is the force balance equation of the liquid cavity: $K$ is the gas stiffness coefficient; $C$ is the gas damping coefficient; and $B$ is the viscous damping coefficient of the oil. $V$ is the gas volume. The above formulas obtain the changing curve of internal pressure and the gas cavity volume when the external pressure, $p_b$, changes. This helps to analyze the dynamic characteristics of the accumulator when it absorbs the pressure pulse.

The volt–current condenser (VCC) servo control loop card model and the unload value system model are shown in Figs 3 and 4, respectively. Each servo system has a control loop card— the VCC module controls the whole system. The default value differs from the current value when modeling. The obtained signal is transformed and processed by Proportional Integral Derivative (PID) control. The unloading value needs a safe oil pressure to prevent the oil pressure in the lower chamber of the oil engine from being too high.

Table 1 shows the model parameters [21].

## Model performance

The system was tested to verify the performance of the DEH system simulation model. A 20mm displacement analog signal was input. Then, the data of each subsystem were collected. The test results are shown in Fig 5A. It can be seen that, when a 20mm displacement signal is input, the current and output oil increase rapidly, and the displacement of the oil motor rises rapidly. When the displacement is close to 20mm, the current and oil output decrease, and the displacement speed of the oil motor slows down [22].

Fig 5B, 5D and 5G shows the VCC control current, servo valve oil output, and oil pump discharge, respectively. It can be seen that when the displacement reaches 20mm they gradually decrease from the peak value to 0, and the downward trend is from high to low. Fig 5F shows

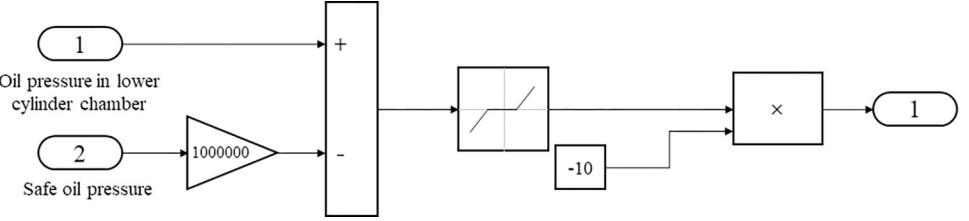

**Fig 4. Unload value system model.**

**Table 1. Model parameters.**

| $A_h$ | $K$ | $K_q$ | $K_f$ | $C_d$ | $M_p$ | $P_0$ | $n$ | $C$ | $R$ | $\rho$ | $\omega$ | $\delta$ |
|---|---|---|---|---|---|---|---|---|---|---|---|---|
| 0.00302 | 300 | 5.084 | 0.382 | 0.02 | 15 | 14.5 | 1500 | 1000 | 0.025 | 855 | 0.016 | 0.7 |

that when the displacement reaches 20mm the oil supply pressure drops steeply and then gradually becomes stable at $1.45 \times 10^7$Pa. The simulation results show that all the subsystems conform to the existing operation law of the DEH system, showing each subsystem's operation status and variation law and laying the foundation for intelligent fault diagnosis and prediction.

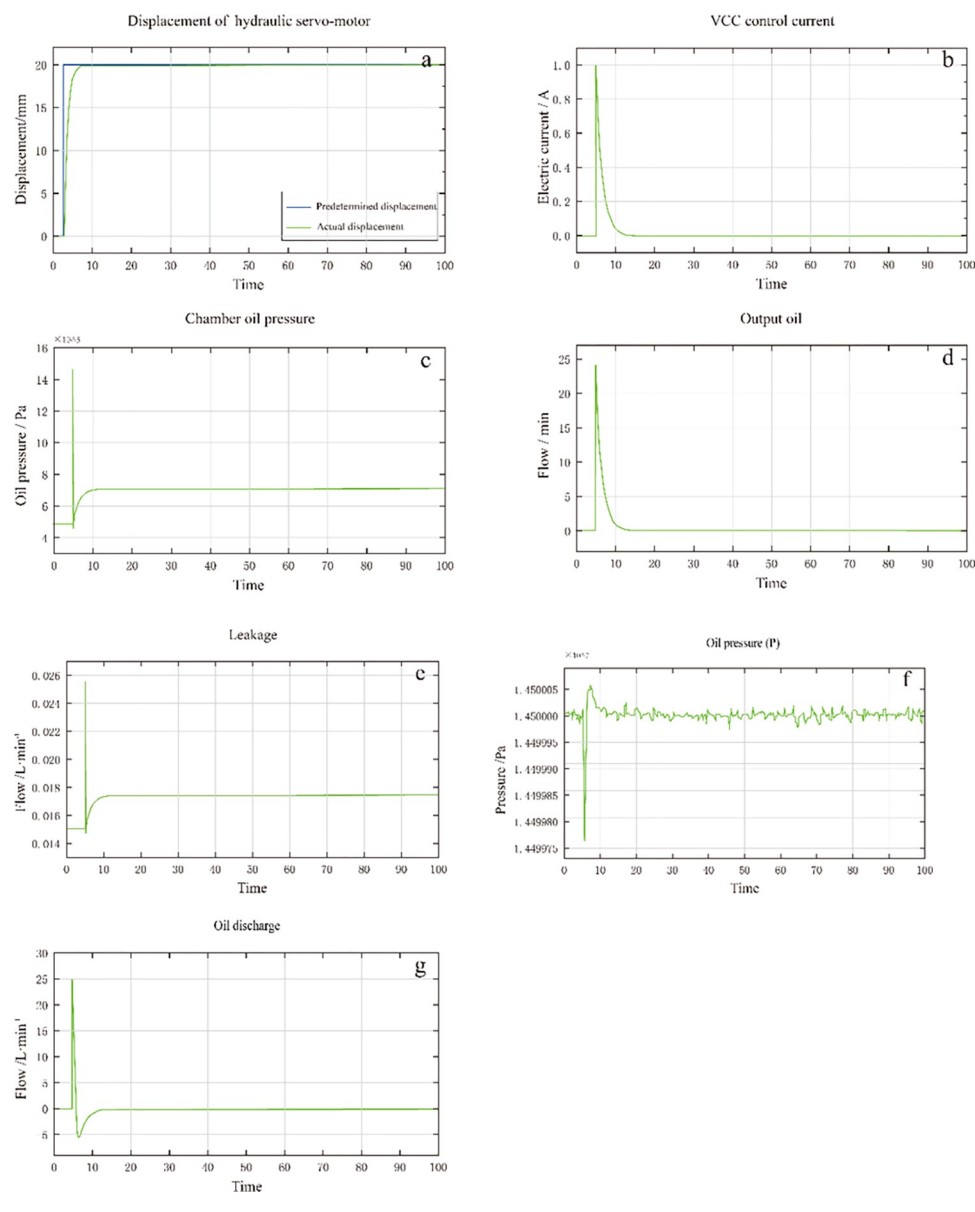

**Fig 5. System simulation results.**

**Table 2. Main faults of the DEH system.**

| Fault type | Electro-hydraulic transducer | Hydraulic servo-motor | Oil system |
|---|---|---|---|
| Jam fault | The armature is stuck or confined; magnetic anomalies; nozzle or orifice is partially blocked | The piston drive is stuck | Deterioration of oil quality leads to mechanical wear |
| Internal leakage fault | Servo valve leakage | Piston wear; plunger and cylinder wear | Oil passage leakage; suction pipe leakage; accumulator leakage; unloading valve leakage |
| Device fault | - | - | The oil cooler is faulty; the accumulator is not fully used; the oil pump is faulty |

## Fault diagnosis and prediction

### Fault type and characteristic parameters

The main parts of the DEH system failure are the electronic control device, EH oil system, actuator, and protection system. Failures of electronic control devices are relatively rare due to the reliability of electronic components and computers. Most faults are caused by the host's communication failure and the card's insensitive contact. It is difficult to judge and eliminate such faults. In addition, due to the influence of the operating environment and other factors, the hydraulic components' reliability is low, so the actuator and the EH oil system covering the hydraulic components become the link to multiple failures. The close cooperation between the protection and EH oil systems cause the failures. Once the EH oil system fails, the protection system will also fail. Therefore, the actuator and EH oil system are the keys to possible failure. Table 2 shows the main faults of the DEH system according to the different components.

According to the system structure and operation characteristics, the principal components causing different faults were deconstructed. The parameters of each component used for fault diagnosis and prediction are shown in Table 3.

### Jam fault

Either the servo valve or the oil motor will cause a jam fault. The main fault that occurs is the failure of the electro-hydraulic converter. Due to the component's strength and other factors, the probability of failure is minimal, but malfunction occurs occasionally. The failure of any of these components will cause a jam fault [23]. Under normal circumstances, the hydraulic servo-motor and the slide valve in the electro-hydraulic transducer are one-to-one relatives. A

**Table 3. Principal components and parameters of the DEH system.**

| Component name | Parameter |
|---|---|
| Electro-hydraulic transducer | Working current of the electro-hydraulic converter ($I_d$) |
| | Pressure difference between inlet and outlet of electro-hydraulic converter ($\Delta p$) |
| Hydraulic servo-motor | Oil pressure of piston upper and lower chambers ($P_{up}$ and $P_{down}$) |
| | Piston displacement signal of oil motor ($x$) |
| Control valve and stem | Valve flow deviation ($\Delta q/q_0$) |
| | Power decay time ($t_r/t_s$) |
| Oil pump | Steam header pressure ($P$) |
| | Current ($I$) |
| | Temperature ($T$) |
| Oil reservoir | Height of liquid level ($h$) |
| Accumulator | Nitrogen pressure ($P_N$) |
| Oil cooler | Working current of the oil cooler ($I_L$) |

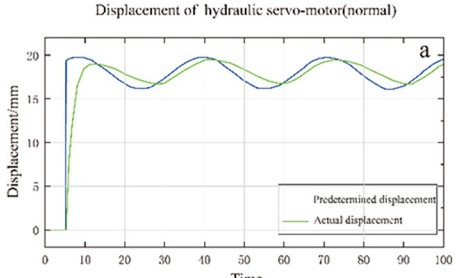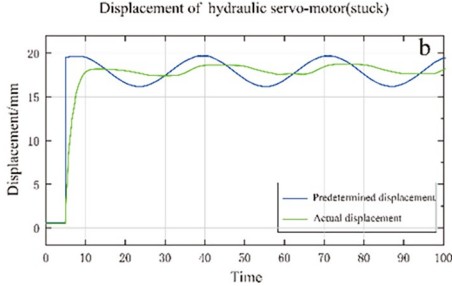

**Fig 6. Displacement comparison of hydraulic servo-motor when servo valve is stuck.**

single slide valve corresponds to a single servo control loop. Due to the influence of nonlinear factors in the actual regulating system, there is an insensitive region in the regulating system: a dead zone. Therefore, the jam fault of servo valve variation can be simulated by adjusting the displacement dead zone in the servo valve module.

When the range of the insensitive area exceeds the range allowed by the typical dead zone, the fault will be manifested in the hydraulic servo-motor and the slide valve becomes stuck. Furthermore, when the insensitive zone far exceeds the dead zone, it is a system failure [24]. The location and extent of the system failure could be diagnosed by judging this range. As shown in Fig 6, when the dead zone is small, the hydraulic servo-motor responds quickly, and the displacement response is closer to the shape of the input signal curve. When the dead zone is larger, the response of the hydraulic servo-motor is slower, and the response curve is closer to a straight line. The servo valve response is not sensitive due to the large dead zone, which leads to the hydraulic servo-motor displacement response not being sensitive. If the dead zone is gradually increased, there will be no response to small sinusoidal signals. This simulation result is consistent with the results appearing on the 300MW steam turbine system [25]. The excellent nature and characteristics of SVM are applied in many areas. Even identifying financial time sequence prediction models [26].

As shown in Fig 7, from the perspective of the VCC card current, when the servo valve is stuck, the current fluctuation amplitude is more significant, which is in line with the actual situation, indicating that the simulation of the jam fault met the research needs.

Fig 8 compares the oil output of the servo valve when the servo valve is stuck. Fig 8A shows the normal state, and Fig 8B shows the jammed state. It can be seen that when the servo valve is stuck, there will be no oil output for part of the time, which is consistent with the actual situation where the servo valve is stuck, resulting in no oil output.

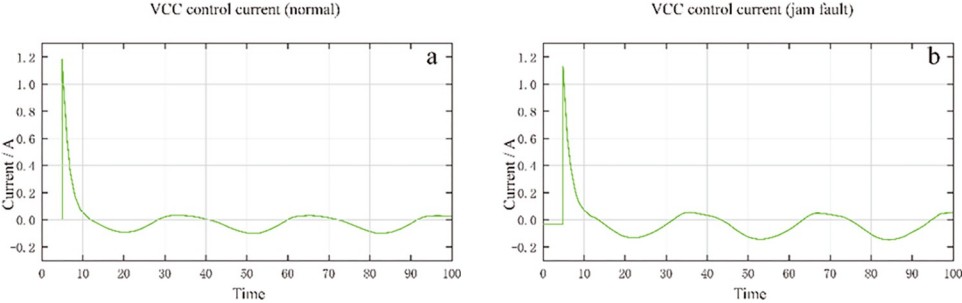

**Fig 7. Comparison of VCC card output current under servo valve jam fault.**

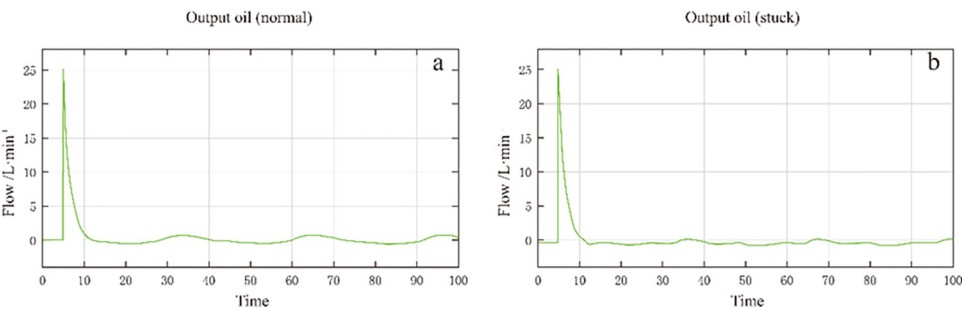

**Fig 8. Comparison of oil output of servo valve when servo valve is stuck.**

## Internal Leakage fault

The leading causes of internal leakage in the DEH system include improper assembly, wear, oil corrosion, etc. When the internal leakage fault occurs in the control system, it will cause corrosion of the spool and the sharp edge of the valve sleeve, accompanied by an increase in oil temperature and volume loss, further reducing the system's reliability.

The internal leakage fault was simulated in the hydraulic servo-motor and servo valve. First, in the hydraulic servo-motor module, the oil pressure in the cavity was multiplied by a leakage coefficient to obtain the leakage amount. Then the leakage amount and the output oil amount of the servo valve were added to obtain the input oil motor's actual amount. A negative bias was added to the output oil of the servo valve in the electro-hydraulic servo valve module to simulate internal leakage. As shown in the displacement change curve of the hydraulic servo-motor in Fig 9, when the leakage coefficient of the hydraulic servo-motor is significant, the displacement response of the hydraulic servo-motor is slow. The current curve shows that the output current is higher when the leakage is more considerable. As shown in Fig 9C, the oil

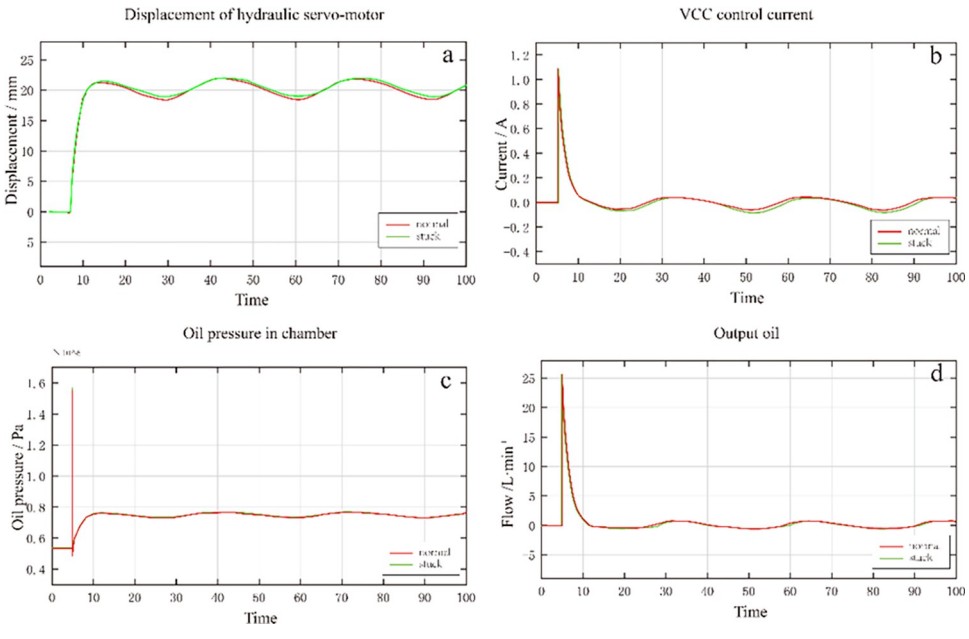

**Fig 9. Comparison of leakage.**

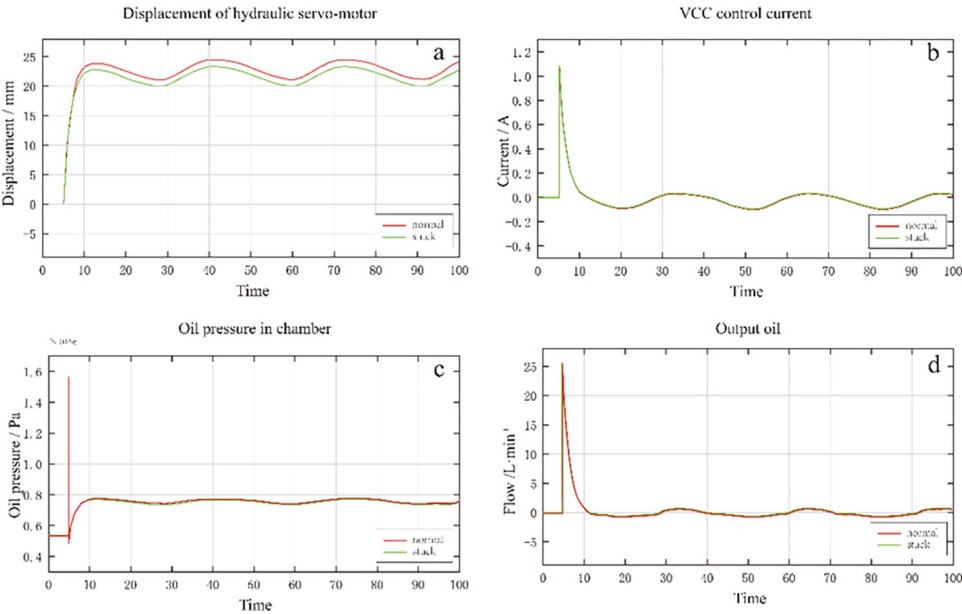

**Fig 10. Comparison of parameters when the LVDT fails.**

pressure in the oil engine chamber increases, but the oil pressure is relatively low when the internal leakage is considerable.

## LVDT feedback fault

The primary function of the linear variable displacement transducer (LVDT) is to collect hydraulic servo-motor displacement information and feedback to the control module [27]. The leading causes of LVDT feedback failure include the ambient temperature, whether the wiring is reliable, and whether the installation is compliant. These causes can cause the internal components to loosen or wear, causing them to feedback incorrect hydraulic servo-motor displacement information.

As the specific principle of the LVDT device was not simulated separately in the model, the displacement feedback of the hydraulic servo-motor was multiplied by a constant, A, in the fault simulation to simulate the overall impact on the system when the LVDT feeds back the wrong displacement of the hydraulic servo-motor. As Fig 10 shows, A was set to 0.95. When the LVDT feedback value is small, the displacement of the oil motor is more significant, and the oil pressure is abnormally high during the fault, but the VCC control current and the oil output of the servo valve do not change much.

## Fault diagnosis model

Using the DEH system model to study the stuck-down faults, multiple faults can be diagnosed and the fault degree can be quantified. Establishing an accurate diagnostic model for the type of oil leakage fault is difficult. The fault diagnosis method based on the data method does not need to establish a system model suitable for simulating the complex nonlinear system. Therefore, the model-based method is used to diagnose stuck faults, and the data-based method is used to diagnose oil leakage faults [28].

**Jam fault diagnosis model.** The stuck fault was simulated by setting the nonlinear dead zone link, as shown in Fig 11.

**Fig 11. Jam fault diagnosis model of the DEH system.**

It can be found from the stuck-down fault model of the DEH system that the stuck-down fault can be diagnosed by identifying the dead zone. By determining the degree and coordinates of the nonlinear dead zone in the system, the degree and location of the stalling fault can be diagnosed [29]. Under normal circumstances, the model output should be the same as the output of the existing system; otherwise, the model structure or parameters should be optimized. There is a difference between the model and the actual output. The difference is called the "residual generator". The system is standard when the residual is less than the set threshold. When the residual exceeds the threshold, the system is faulty. The analysis of residual signals makes it possible to diagnose system faults.

**Leakage fault diagnosis model.** The leakage fault diagnosis model can be divided into data acquisition and preprocessing, robust fault feature extraction, and diagnosis model construction [30]. The DEH system simulation model collects data in normal and fault states. The collected data were preprocessed by zero-mean unit variance (Z-Score) in different dimensions to obtain $\bar{X}$. The formula for data preprocessing is as follows:

$$\bar{X} = \frac{X - mean(X)}{std(X)} \tag{11}$$

*where mean(X) and X are vectors of the mean and standard deviation of features per dimension, respectively.*

The model was divided into two parts in robust fault feature extraction, as shown in Fig 12. In the first part, a two-layer principal component analysis (PCA) model was used as a data processing layer to process and decompose data. The second part used principal component analysis (PCA1~4) and independent component correlation algorithm (ICA1~4) models as feature extraction layers. The PCA model was used to extract Gaussian features from the data, and the ICA model was used to extract non-Gaussian features from the data.

The amount of DEH system fault diagnosis data was small. The intention of this research was to build a multi-classification fault diagnosis model through the small-sample learning method to achieve a good fault diagnosis effect. The Support Vector Machine (SVM) shows excellent stability and anti-interference when dealing with small samples [31].

The parameters were determined by cross-validation of the training data. For the sample data of a DEH system, the fault diagnosis model will make the fault diagnosis conclusion by the SVM according to the fseature information.

First, the feature information of the training data was used to train the SVM classifier, and the fault type was directly given by the SVM. The objective function of this method is:

$$\min_{a} \frac{1}{2} \sum_{i=1}^{N} \sum_{j=1}^{N} a_i a_j y_i y_j K(x_i, x_j) - \sum_{i=1}^{N} a_i \tag{12}$$

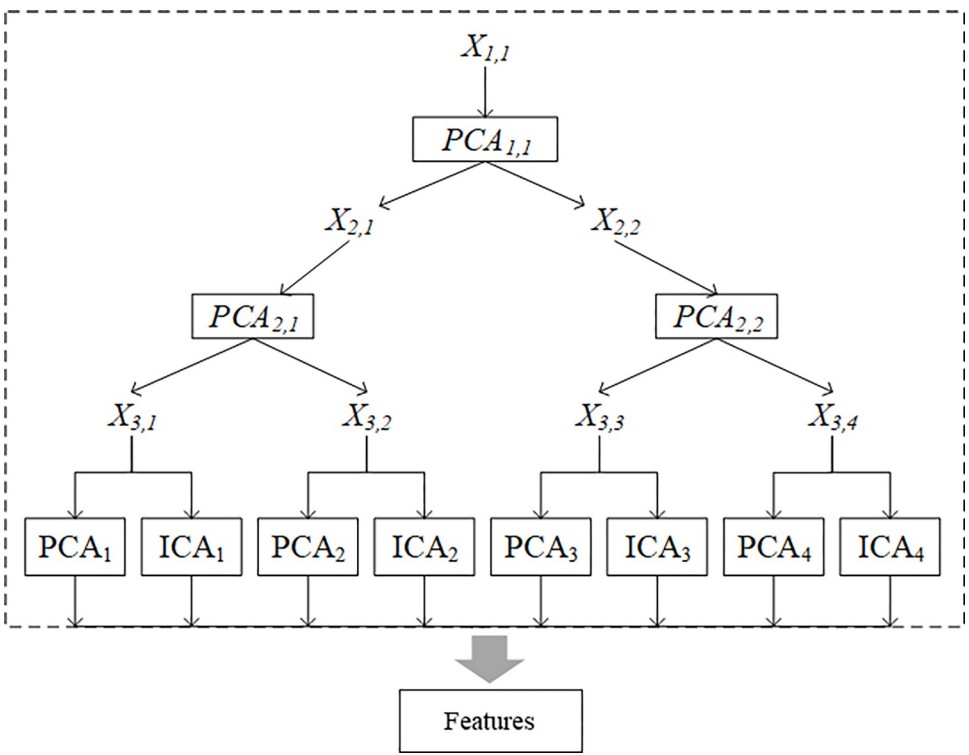

**Fig 12. Robust fault feature extraction model.**

$$s.t \sum_{i=1}^{N} y_i a_i = 0 \tag{13}$$

$$0 \leq a_i \leq C \tag{14}$$

where C is the error cost, which is used to describe the tolerance of the SVM for the classification errors of individual outliers.

$$F(X) = sign\left[\sum_{i=1}^{N} a_i y_i K(X_i, X_i) + b'\right] \tag{15}$$

$$b' = y_j - \sum_{i=1}^{N} a_i y_i K(X_i, X_j) \tag{16}$$

Then, the above steps were repeated for cross-validation with different parameters until a satisfactory parameter value was obtained. Finally, the final fault diagnosis model was utilized to verify the model performance on the independent test set.

## Fault prediction model

The fault prediction model is the basis of predictive maintenance. There are two fault prediction methods: physical model and data-driven method. This study constructed a multi-parameter gray error neural network prediction model with high prediction accuracy and real-time

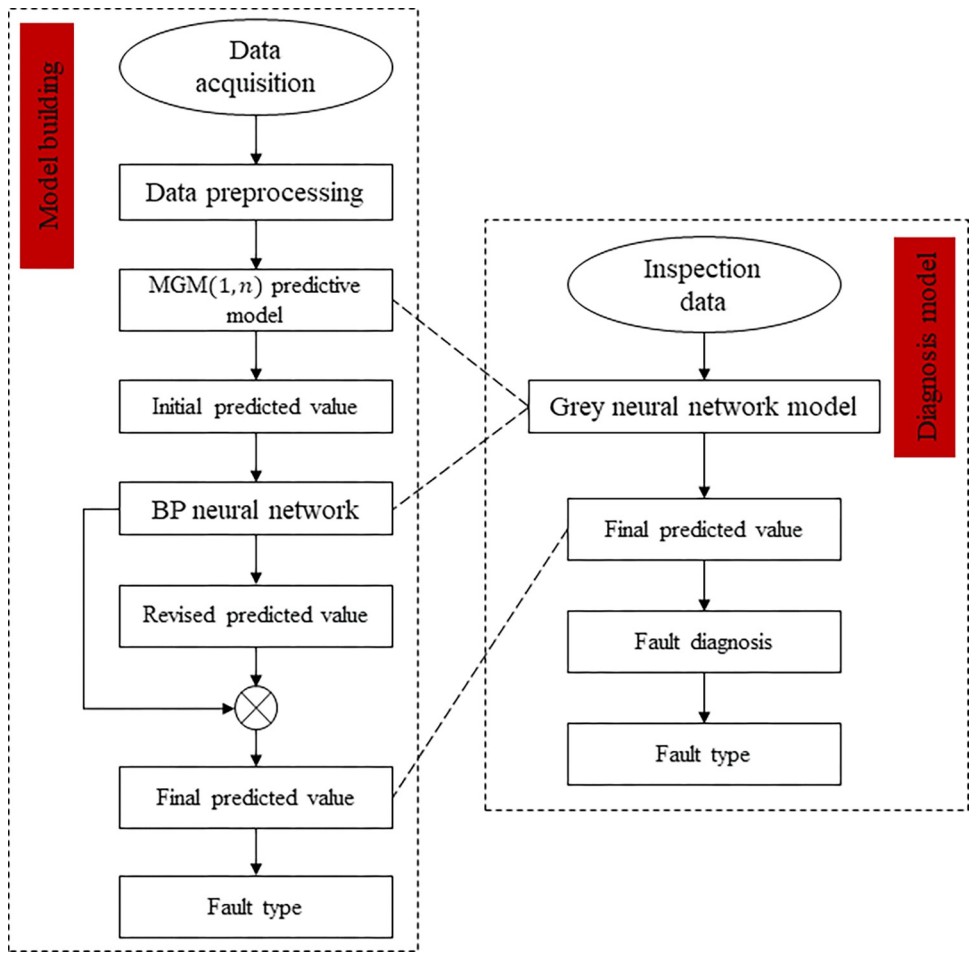

**Fig 13. Flowchart of the fault prediction model.**

performance. As shown in Fig 13, the advantages of the gray system prediction model and the neural network system prediction model are combined to improve the fault prediction accuracy [32].

This research used the predicted value of the gray system prediction model's output as the neural network model's input and the residual between the predicted value and the actual value as the output. The minimum sum of the squared residuals was taken as the training target of the neural network model. A neural network to modify the prediction value of the gray system prediction model can significantly improve the prediction accuracy [33].

**Data acquisition.**   The feature vectors of the turbine DEH system in the period before the fault occurred were recorded to construct the time data series. The data sequence of a single component and single factor are denoted as $\{x(1),x(2),...,x(n)\}$. When any feature value of the component reaches the threshold, $X$, the turbine will be stopped and preventive maintenance or replacement will be carried out. Fault prediction is based on the known data sequence $\{x(1),x(2),...,x(n)\}$, using a specific algorithm to obtain the predicted data sequence $\{x(n+1),x(n+2),...,x(n+m)\}$. The changing trend of data and the correctness of future state data are the basis for evaluating the performance of fault prediction methods. In addition, the prediction's absolute and relative errors can also be analyzed to evaluate the advantages and disadvantages of the prediction model and its adaptability.

**MGM(1,$n$) (Multi-variablegrey Model) predictive model [34].** Suppose there are $n$ characteristic indicators, and the original time sequence of the $i$th indicator is as follows:

$$X_i^{(0)} = \{x_i^{(0)}(k)\}(1 \le i \le n, 1 \le k \le m). \tag{17}$$

The monotone sequence was generated by accumulating them once as follows:

$$X_i^{(1)} = \{x_i^{(1)}(k)\}(1 \le i \le n, 1 \le k \le m), \ where, \ x_i^{(1)}(k) = \sum_{j=1}^{k} x_i^{(0)}(j) \tag{18}$$

$X^{(i)} = (X_1^{(i)}, X_2^{(1)}, \ldots, X_n^{(1)})(0 \ge i \le 1)$ was recorded. A system of an n-ary first-order differential equation was established for the sequence $X^{(1)}$ generated by one-time accumulation, which is expressed by a matrix as follows:

$$\frac{dX^{(1)}}{dt} = AX^{(1)} + U \tag{19}$$

The equation set is called the winterization equation of the MGM(1,n) model, where $A$ is the development coefficient matrix, $A = (a_{ij})_{n\times n}$ $(i,j = 1,2,\ldots,n)$, and $U$ is the gray function matrix.

The prediction accuracy of MGM(1,n) depends on the parameter matrix, $A$, and the gray function matrix, $U$. A and U depend on the original sequence and the constructed form of the background values. Generally, the adjacent mean of $X^{(1)}$ is used to generate sequence values as background values, $Z^{(1)}(k)$:

$$Z^{(1)}(k) = 0.5[X^{(1)}(k) + X^{(1)}(k-1)] \tag{20}$$

The basic form of the MGM(1,n) prediction model is obtained by replacing $X^{(1)}(k)$ of the whitening equations $Z^{(1)}(k)$:

$$X^{(0)}(k) = AZ^{(1)}(k) + U, \ k = 2, 3, \ldots, m. \tag{21}$$

The estimated values of parameters $A$ and $U$ can be obtained by least squares.

$$\hat{a}_i = (L^T L)^{-1} L^T Y_i, \ a_i = (a_{i1}, a_{i2}, \ldots, a_{in}, u_i)^T, \ i = 1, 2, \ldots, n \tag{22}$$

$$where \ L_j = \begin{pmatrix} 0.5(x_j^{(1)}(2) + x_j^{(1)}(1)) & 1 \\ 0.5(x_j^{(1)}(3) + x_j^{(1)}(2)) & 1 \\ \vdots & \vdots \\ 0.5(x_j^{(1)}(m) + x_j^{(1)}(m-1))1 \end{pmatrix}, \ Y_i = (x_i^{(1)}(2), x_i^{(1)}(3), \ldots, x_i^{(1)}(m))^T.$$

If $\hat{X}^{(0)}(1) = X^{(0)}(1) = X^{(1)}(1)$, then the continuous-time response function of the system of differential equations is as follows:

$$X^{(1)}(t) = e^{At} X^{(1)}(0) + A^{-1}(e^{At} - E)U \tag{23}$$

where $e^{At} = E + At + \frac{A^2}{2!}t^2 + \ldots + \frac{A^n}{n!}t^n + \ldots = E + \sum_{k=1}^{\infty} \frac{A^k}{k!}t^k$. $E$ is the identity matrix. The prediction value of the MGM(1,n) model can be obtained by discretizing the equation above:

$$\hat{X}^{(1)}(k) = e^{\hat{A}(k-1)} X^{(1)}(1) + \hat{A}^{-1}(e^{A(k-1)} - E)\hat{U}, \ k = 1, 2, \ldots. \tag{24}$$

The predicted value of the original sequence can be obtained by the following formula of the reduction operator:

$$\hat{X}^{(0)}(k) = \hat{X}^{(1)}(k) - \hat{X}^{(1)}(k-1), \ k = 2, 3, \ldots. \tag{25}$$

Suppose $X^{(0)}(k) = (x_1^{(0)}(k), x_2^{(0)}(k), \ldots, x_n^{(0)}(k))$, $1 \leq k \leq m$ is the original value vector of all characteristic indices that affect the faulty system at time $k$. Through the MGM(1,n) prediction model, the predicted value vectors of $n$ feature indexes $\hat{X}^{(0)}(k)$ and $\hat{X}^{(0)}(k+1)$ at time $k$ and time $k+1$ were obtained. The error value vector, $E(k)$, between the predicted value and the actual value of $n$ characteristic indexes at time $k$ was obtained:

$$\hat{X}^{(0)}(k) = (\hat{x}_1^{(0)}(k), \hat{x}_2^{(0)}(k), \ldots, \hat{x}_n^{(0)}(k)) \tag{26}$$

$$\hat{X}^{(0)}(k+1) = (\hat{x}_1^{(0)}(k+1), \hat{x}_2^{(0)}(k+1), \ldots, \hat{x}_n^{(0)}(k+1)) \tag{27}$$

$$E(k) = (e_1(k), e_2(k), \ldots, e_n(k)),$$
$$(e_i(k) = x_i^{(0)}(k) - \hat{x}_i^{(0)}(k), 1 \leq i \leq n, 1 \leq k \leq m) \tag{28}$$

**BP neural network to correct the predicted value.** A three-layer BP neural network was used to correct the MGM(1,n) error term by combining the serial and embedded gray neural network hybrid models. Then, the predicted value $\hat{X}^{(0)}(1), \hat{X}^{(0)}(2), \ldots, \hat{X}^{(0)}(m)$ of the original sequence was used as the input of the neural network [35]. $E(1),E(2),\ldots,E(m)$ was used as the corresponding network expected output.

Output of hidden layer nodes:

$$\hat{y}_j = f(\sum_i w_{ji}\hat{x}_i - \theta_j) \tag{29}$$

The output layer nodes:

$$z_l = f(\sum_j v_{lj}\hat{y}_j - \theta_l) \tag{30}$$

*where* $f(x) = (1 - e^{-2x})/(1 + e^{-2x})$.

The process of training neural networks is essential to finding a set of network weights ($w_{ji}$, $v_{lj}$) and threshold values ($\theta_j$, $\theta_l$) that minimize the corresponding network error function. The error function is as follows:

$$E = \frac{1}{2}\sum_i(e_l - z_l)^2 = \frac{1}{2}\sum_i(e_l - f(\sum_j v_{lj}\hat{y}_j - \theta_l))^2$$
$$= \frac{1}{2}\sum_i(e_l - f(\sum_j v_{lj}f(\sum_i w_{ji}\hat{x}_i - \theta_j) - \theta_l))^2 \tag{31}$$

*where* $e_l$ is the expected output of the network and $z_l$ is the actual output.

The weight was modified as:

$$w_{ji}(k+1) = w_{ji}(k) + \Delta w_{ji}(k) + \beta \Delta w_{ji}(k) \tag{32}$$

$$v_{lj}(k+1) = v_{lj}(k) + \Delta v_{lj}(k) + \beta \Delta v_{lj}(k) \tag{33}$$

*where* $\Delta w_{ji}$ and $\Delta v_{lj}$ are the adjustment quantities of the BP algorithm; $\beta\Delta w_{ji}(k)$ and $\beta\Delta v_{lj}(k)$ are the momentums; and $\beta$ is the momentum coefficient.

Then, the trained network was used for the prediction simulation. Vector $\hat{X}^{(0)}(m+1)$ was taken as the network input vector. After the trained network calculation, the output of the network is the error correction predicted value vector of $n$ eigenvalue indicators at time $m+1$, $E(m+1) = (e_1(m+1), e_2(m+1), \ldots, e_n(m+1))$.

The actual predicted value vector $X^*(m+1)$ of $n$ eigenvalue indicators at time $m+1$ is the sum of the predicted value vector of the gray system $\hat{X}^{(0)}(m+1)$ and the expected value vector of the modified value indicated by the neural network $E(m+1)$.

$$X^*(m+1) = \hat{X}^{(0)}(m+1) + E(m+1) \tag{34}$$

$$where \ \ x_i^*(m+1) = \hat{x}_i^{(0)}(m+1) + e_i(m+1), (1 \leq i \leq n)$$

$$X^*(m+1) = (x_1^*(m+1), x_2^*(m+1), \ldots, x_n^*(m+1))$$

$$\hat{X}^{(0)}(m+1) = (\hat{x}_1^{(0)}(m+1), \hat{x}_2^{(0)}(m+1), \ldots, \hat{x}_n^{(0)}(m+1))$$

$$E(m+1) = (e_1(m+1), e_2(m+1), \ldots, e_n(m+1)).$$

This kind of forecasting model is a dynamic rolling forecasting model. The information with a small correlation was removed and new information was injected into the prediction model to make the predicted value closer to the actual value. This improves prediction accuracy and provides a more scientific basis for decision making [36].

## The embedded platform

We developed an embedded platform that uses the TensorFlow framework to mount trained models onto a dedicated integrated circuit chip. The embedded platform utilizes various types of sensors to collect pressure data, pH data, flow data, temperature data, and displacement data from various parts of the turbine. We will transmit the collected data to the fault diagnosis and prediction model to determine whether the DEH system is functioning properly and then predict that it will fail at a certain time.

As shown in Fig 14, We started with the Edge TPU, which is a smart chip with an arithmetic power of 4 TOPS. We configured the runtime environment needed for the Edge TPU. Next, we installed the Tensorflow framework, and other dependent packages. Next, the configuration files were modified and compiled. Once this was done, we put the system code on the Edge TPU and ran it to diagnose and predict system failures. This embedded platform performs faster and with less loss of accuracy than the previous control.

## Conclusions

By analyzing the DEH system's actual running condition, the DEH system simulation model was built by SIMULINK tool, and the model's performance was tested. Based on the simulation model of the DEH system, fault simulation was carried out, and the fault diagnosis and prediction model was built.

(1) Physical object-oriented modeling was adopted, and the simulation model of the DEH system was very close to the actual working situation of the system. Compared with the existing method of establishing a model, the method selected in this study is simpler and easy to achieve.

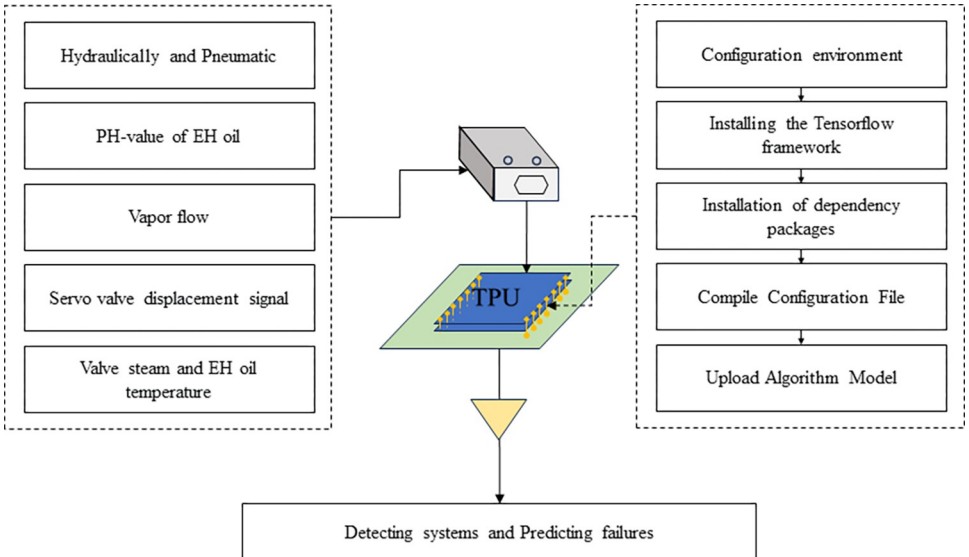

**Fig 14. Intervention process of the embedded platform on the system.**

The displacement of the oil engine, the oil pressure of the oil engine chamber, the oil supply pressure, and the oil discharge of the oil pump all conformed to the operation law of the steam turbine, which indicates that the model meets the requirements of simulation and research.

(2) A model-based method was used to identify the card astringency fault, and the data-based method was used to diagnose the leakage fault. The fault data collection and analysis show that the model can effectively identify the faults of the DEH system. The fault diagnostic model can almost simulate all the types of faults that occur in actual work, and it can diagnose unknown failure types.

(3) A multi-parameter gray error neural network prediction model was constructed by combining the advantages of the neural network and the gray system. The predicted value obtained from the gray system prediction model was taken as the input of the neural network, and the error between the expected value and the actual value was taken as the neural network's output.

In the next work, we will further study the actual effect of the DEH system failure prediction of the turbine DEH system. Use the prediction model to determine the time it can know in advance before the fault.

## Author Contributions

**Conceptualization:** Ling Zhong, Qing Li.

**Formal analysis:** Qing Li.

**Methodology:** Qing Li.

**Writing – original draft:** Ling Zhong.

**Writing – review & editing:** Ling Zhong.

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
