## [Decision Letter · Decision Letter 0]

21 Jun 2023

PONE-D-23-15933Intelligent Diagnosis and Prediction of Turbine DEH System Faults: Design and ExperimentationPLOS ONE

Dear Dr. Zhong,

Thank you for submitting your manuscript to PLOS ONE. After careful consideration, we feel that it has merit but does not fully meet PLOS ONE’s publication criteria as it currently stands. Therefore, we invite you to submit a revised version of the manuscript that addresses the points raised during the review process.

ACADEMIC EDITOR: Major Revision

We look forward to receiving your revised manuscript.

Kind regards,

Aytaç Altan, Ph.D.

Academic Editor

PLOS ONE

Journal Requirements:

"NO authors have competing interests."

Additional Editor Comments:

Dear Authors,

The reviewers have evaluated your manuscript and suggested that certain aspects be approached with caution. To address their feedback, it is recommended that you make revisions to your manuscript. Additionally, along with the reviewers' comments, it would be beneficial to carefully consider the following points:

1) It is important to analyze the computational complexity of the proposed algorithm.

2) How can the parameters of the proposed method be set to achieve better performance?

3) The paper lacks a clear explanation of the significance of the design compared to other notable works in the field. The authors should review, comment on, and compare more recent studies. It is essential to emphasize the study's contribution and ensure that the references are up-to-date. The following studies can be referenced to elucidate the proposed process: " The effect of kernel values in support vector machine to forecasting performance of financial time series and cognitive decision making" and " A new hybrid model for wind speed forecasting combining long short-term memory neural network, decomposition methods and grey wolf optimizer".

Reviewers' comments:

Reviewer's Responses to Questions

**Comments to the Author**

1. Is the manuscript technically sound, and do the data support the conclusions?

Reviewer #1: Yes

Reviewer #2: Yes

2. Has the statistical analysis been performed appropriately and rigorously? 

Reviewer #1: Yes

Reviewer #2: Yes

3. Have the authors made all data underlying the findings in their manuscript fully available?

Reviewer #1: Yes

Reviewer #2: Yes

4. Is the manuscript presented in an intelligible fashion and written in standard English?

Reviewer #1: Yes

Reviewer #2: Yes

5. Review Comments to the Author

Reviewer #1: The manuscript is a good contribution to the field of engineering, and it has brought considerable knowledge on the subject. The author must add considerable number of citations in the main text to support all the results and discussion made. Graphs and pictures should be made clearer and more readable. Comparisons on the proposed systems with the previous models need to be based on the statistical analysis. Would there be a need to use MSD (minimal statistical differences) for comparisons to be based on? Or find similar tests which are relevant to the study.

Reviewer #2: 1. The title should be modified to not include any abbreviation. Define DEH in the title to be clear to the readers.

2. The abstract is long and needs to be rewritten in compact form. In general, try to shorten the ideas, simplify the content of the article, and make an adequate order of the ideas. The abstract needs to be written in a more clarified way indicating the contribution and the superior of one proposed method over another.

3. Survey of existing literature is not sufficient. It would useful to include in the Introduction of the paper some discussion on other possible real applications of the obtained results and at the end of the literature, the weakness of previous research has to be highlighted.

4. The author needs to give some practical examples to support the work.

5. What are the main contributions of the paper compared with other previous related works? What is the new in your study? The contribution is not clearly indicated. The author highlighted the conclusion at the end of introduction, not a contribution. The contribution has to be stated in points.

6. A brief description of the structure of the paper should be added at the end of the introduction section.

7. The format and English writing of this paper should be improved.

8. Some abbreviations need to check when writing for the first time, like DEH, AI, SVM, MGM, and PID.

9. The equations are badly written and edited, see eq. 1, eq. 3, eq. 5, eq. 9, eq. 10, eq. 12, eq. 13, eq. 14, eq. 18, eq. 28,

10. Some notations in equations have not been defined, see eq. 1, eq. 3, eq. 6, eq. 7, eq. 9, and eq. 10.

11. The article is badly presented, organized and analyzed.

12. The analysis and presentation of fault diagnosis and prediction section is not rigorous and not clear and it needs to rewritten in more clear way.

13. The article is mainly developed according to other research, which is spread alongside the manuscript. Where is the novelty?

14. The block diagram of controlled system has to be put to explain the mechanism of control. Figure (2) does no show the whole control scenario. Discussion for it is also recommended.

15. The choice of design parameters is still a challenging problem. If not possible to perform one of the modern techniques in tuning these design parameters, I suggest referring to some of them.

16. References [29] are not cited in the text.

17. In the simulation results section, the proposed approaches should compare with other relevant techniques which cited in the literature reviews. The authors are asked to further provide comparisons to better show the advantages of their approaches.

18. The discussion of simulation is not based on reasoning.

19.The deep discussion is required in simulated results.

20. The conclusion is descriptive. It is void of quantitative and numerical improvement and comparison and the conclusion has to be extended with future work.

21. The study did not address the practical verification of simulated results. The work is still weak unless it is experimentally implemented.

6. PLOS authors have the option to publish the peer review history of their article (what does this mean?). If published, this will include your full peer review and any attached files.

Reviewer #1: No

Reviewer #2: No

---

## [Author Response · Author response to Decision Letter 0]

17 Aug 2023

2 Review Comments to the Author

Reviewer #1

[Comment]

The manuscript is a good contribution to the field of engineering, and it has brought considerable knowledge on the subject. The author must add considerable number of citations in the main text to support all the results and discussion made. Graphs and pictures should be made clearer and more readable. Comparisons on the proposed systems with the previous models need to be based on the statistical analysis. Would there be a need to use MSD (minimal statistical differences) for comparisons to be based on? Or find similar tests which are relevant to the study.

[Response for Reviewer #1]

Thanks for bringing up these critical points. Numerous citations have been added to this thesis to support the conclusions and discussions obtained. Graphs and images have been optimized and the original vector graphics have been provided for clarity and readability. The model presented in this thesis is not compared with previous models, but with actual situations in the section titled "Model performance". The failure prediction model is for diagnosis and prediction of DEH system failures, and similar tests could not be found to compare and analyze the model with MSD, but a description of the previous research model is added in the text to support the accuracy of the model proposed in this thesis.

Reviewer #2

[Comment #1]

The title should be modified to not include any abbreviation. Define DEH in the title to be clear to the readers.

[Response #1]

We truly appreciate the reviewer’s valuable comments. According to the suggestions of reviewers, we have revised the title of this manuscript. The revised title was as follows.

Intelligent Diagnosis and Prediction of Turbine Digital Electro-Hydraulic Control System Faults: Design and Experimentation.

[Comment #2]

The abstract is long and needs to be rewritten in compact form. In general, try to shorten the ideas, simplify the content of the article, and make an adequate order of the ideas. The abstract needs to be written in a more clarified way indicating the contribution and the superior of one proposed method over another.

[Response #2]

The authors thank the reviewer for this comment. The abstract section of the study was streamlined and optimized to ensure brevity and accuracy in the content of the abstract. 

[Line 9-22, Page 2] Abstract

A physical modeling approach was adopted to build a Digital Electro-Hydraulic Control (DEH) system simulation model and the fault models using the SIMULINK tool. This research combined the advantages of the gray system and neural network to build a multi-parameter gray error neural network fault prediction model for the first time. Furthermore, an embedded platform for intelligent fault diagnosis and prediction was developed using an Application Specific Integrated Circuit chip. The results show that the simulation model of the DEH system has good performance. A jam fault, internal leakage, and a device fault could be accurately identified through the fault diagnosis model. The multi-parameter gray error neural network prediction model improves the accuracy of fault prediction. The embedded platform developed by the Application Specific Integrated Circuit chip solves the problem of transmission limitation and insufficient computing power. It realizes the intelligent diagnosis and prediction of DEH system faults and guarantees the regular operation of the DEH system.

[Comment #3]

Survey of existing literature is not sufficient. It would useful to include in the Introduction of the paper some discussion on other possible real applications of the obtained results and at the end of the literature, the weakness of previous research has to be highlighted. 

[Response #3]

We have revised the introduction to improve the current research and so as the references.

[Line 71-79, Page 5] Introduction

The research methods and types of other scholars are not rich, and there are many shortcomings. They have done some work, which involves collecting data on faults that have occurred and analyzing the causes of the faults [13]. But their fault diagnosis methods mostly diagnose vibration faults of steam turbines, and even this diagnostic tool becomes helpless when encountering unknown types of faults [14]. Other scholars' research tools did not take the Mega Data into account, and did not use the collected data to predict the occurrence of failures and prevent them. The first step, we build a DEH system simulation model using the SIMULINK tool.

[Line 432-444, 451-453, 465-469, 477-484, 488-490, 494-501, 522-524, 528-530, Page 30-34] Reference

[2] WANG, SHENG-HUI, NI, TONG-WEI, YANG, ZHEN-GUO. Failure analysis on abnormal blockage of electro-hydraulic servo valve in digital electric hydraulic control system of 125 MW thermal power plant[J]. Engineering failure analysis,2021,123. 

[3] Graciano D M, Rodríguez J A, Urquiza G, et al. Damage evaluation and life assessment of steam turbine blades[J]. Theoretical and Applied Fracture Mechanics, 2023, 124: 103782.

[4] Karakurt A S, ÖZSARI İ, Başhan V, et al. Evolution of steam turbines: A bibliometric approach[J]. Journal of Thermal Engineering, 2021, 8(5): 681-690.

[5] Liu Y, Yin X, Yi Y, et al. Design, simulation, and experiment research on fast control system of steam turbine inlet valve[J]. Proceedings of the Institution of Mechanical Engineers, Part E: Journal of Process Mechanical Engineering, 2022, 236(2): 452-462.

[8] Altan A, Karasu S, Zio E. A new hybrid model for wind speed forecasting combining long short-term memory neural network, decomposition methods and grey wolf optimizer[J]. Applied Soft Computing, 2021, 100: 106996.

[13] Liang Z, Zhang L, Wang X. A novel intelligent method for Fault Diagnosis of steam turbines based on T-SNE and XGBoost[J]. Algorithms, 2023, 16(2): 98.

[14] Bovsunovsky A, Shtefan E, Peshko V. Modeling of the circumferential crack growth under torsional vibrations of steam turbine shafting[J]. Theoretical and Applied Fracture Mechanics, 2023, 125: 103881.

[18] Zhu J F. Design of an experimental platform for hydraulic servo system[J]. Applied Mechanics and Materials, 2011, 69: 51-54.

[19] Altan A, Karasu S. The effect of kernel values in support vector machine to forecasting performance of financial time series[J]. The Journal of Cognitive Systems, 2019, 4(1): 17-21.

[20] Damasceno M A, Penha J K M, Silva Junior N F, et al. Influence of the temperature, pressure and viscosity on the oil measurement with turbine type measurers[J]. Brazilian Archives of Biology and Technology, 2006, 49: 65-72.

[22] Qi J Y, Jin G, Yang T, et al. Effect of Turbine DEH Proportional Differential Feedforward Control on Stability of the Power System[J]. Journal of Chinese Society of Power Engineering. 2019, 4, 39.

[24] DONG Yu-liang, GU Yu-jiong, ZHANG Yi. Maintenance decision on steam turbine digital electro-hydraulic control system based on risk [C]//Automation and Logistics. Beijing, 2008: 764-768.

[25] Zhang W Q, Zhang Y Y, Analysis on reasons of jam of high-pressure main stop valve in 300MW steam turbine[J]. Huadian Technology, 2011, 07,10-11(79).

[26] Yu B, Zhu Q, Yao J, et al. Design, mathematical modeling and force control for electro-hydraulic servo system with pump-valve compound drive[J]. IEEE Access, 2020, 8: 171988-172005.

[34] Luo Y, Che X, He Z. Non-Equidistance Multivariate Optimization Model MGM (1, n) and Its Application[J]. Electronic Journal of Geotechnical Engineering, 2015, 20: 11189-11196.

[36] Li S, Li G, Wang Z. Fault Prediction of Electromagnetic Brake Based on AINN and Grey MGM (1, n) Model[C]//Journal of Physics: Conference Series. IOP Publishing, 2020, 1626(1): 012010.

[Comment #4]

The author needs to give some practical examples to support the work.

[Response #4]

We add work on how the model can be laid out on an Application Specific Integrated Circuit (ASIC) chip. A practical example of which we will carry out experimental validation in a subsequent study.

[Line 397-403, Page 28] The embedded platform

We developed an embedded platform that uses the TensorFlow framework to mount trained models onto a dedicated integrated circuit chip. The embedded platform utilizes various types of sensors to collect pressure data, pH data, flow data, temperature data, and displacement data from various parts of the turbine. We will transmit the collected data to the fault diagnosis and prediction model to determine whether the DEH system is functioning properly and then predict that it will fail at a certain time.

[Comment #5]

What are the main contributions of the paper compared with other previous related works? What is the new in your study? The contribution is not clearly indicated. The author highlighted the conclusion at the end of introduction, not a contribution. The contribution has to be stated in points.

[Response #5]

We reorganize the main contributions of this research and revised in the Introduction. In this research, a new DEH system is proposed, and a new fault diagnosis model and prediction model for DEH system is built. 

[Line 79-82, Page 5] Introduction

In this paper, by combining data and model. We propose a new method for diagnosing the fault types of DEH systems by extensive data analysis. This can make up for the defects in model analysis, control the component parameter information in the control system in real-time.

[Comment #6]

A brief description of the structure of the paper should be added at the end of the introduction section.

[Response #6]

The authors thank for the comment. We have revised the introduction in a brief description of the structure of the paper.

[Lines 78-86, Page 5] Introduction

The first step, we build a DEH system simulation model using the SIMULINK tool. Then, in this paper, by combining data and model. We propose a new method for diagnosing the fault types of DEH systems by extensive data analysis. This can make up for the defects in model analysis, control the component parameter information in the control system in real-time. Finally, we proposed a prediction model of DEH system failure for the first time. Through this model, we can accurately know the probability of failure in the DEH system of the turbine, and know when it will stop working, and maintain the safe operation of steam turbines and ensure the safety of power production.

[Comment #7]

The format and English writing of this paper should be improved.

[Response #7]

The authors thank for the comment. 

We checked and revised the language of the manuscript. We have had our language edited by a native English speaker. Figure 1* shows the language polish certificate. If there still have any language problems, please do not hesitate to tell us.

Figure 1*. Language polish certificate

[Comment #8]

Some abbreviations need to check when writing for the first time, like DEH, AI, SVM, MGM, and PID.

[Response #8]

Thanks for these constructive suggestions. All abbreviations have been clarified and replaced in the new changes. 

[Lines 9-10, page 2]

Digital Electro-Hydraulic Control (DEH)

[Lines 146, page 9]

volt–current condenser (VCC)

[Lines 285, page 20]

principal component analysis (PCA)

[Lines 332, page 24]

MGM(1,n) (Multi-variablegrey Model) predictive model

[Comment #9]

The equations are badly written and edited, see eq. 1, eq. 3, eq. 5, eq. 9, eq. 10, eq. 12, eq. 13, eq. 14, eq. 18, eq. 28,

[Response #9]

The authors thank you for the comments. We have revised this manuscript and all equations were re-edited to make sure they looked correct. 

[Comment #10]

Some notations in equations have not been defined, see eq. 1, eq. 3, eq. 6, eq. 7, eq. 9, and eq. 10.

[Response #10]

The authors thank you for the comments. As you mentioned in the comment, we have revised this manuscript to make sure the symbols appearing in all equations are defined.

[Comment #11]

The article is badly presented, organized and analyzed.

[Response #11]

According to the suggestions, we have revised the introduction, conclusion and figures in the manuscript to make it present well. 

[Figure 5, page 11-12]

Figure 5. System simulation results

[Figure 6, page 15]

This simulation result is consistent with the results appearing on the 300MW steam turbine system [25]. The excellent nature and characteristics of SVM are applied in many areas. Even identifying financial time sequence prediction models [26].

Figure 6. Displacement comparison of hydraulic servo-motor when servo valve is stuck.

[Figure 7, page 16]

Figure 7. Comparison of VCC card output current under servo valve jam fault.

[Figure 8, page 16]

Figure 8. Comparison of oil output of servo valve when servo valve is stuck.

[Figure 9, page 17]

Figure 9. Comparison of leakage.

[Figure 10, page 18-19]

Figure 10. Comparison of parameters when the LVDT fails.

[Line 404-427, page 28-29] Conclusions 

By analyzing the DEH system's actual running condition, the DEH system simulation model was built by SIMULINK tool, and the model's performance was tested. Based on the simulation model of the DEH system, fault simulation was carried out, and the fault diagnosis and prediction model was built. 

(1) Physical object-oriented modeling was adopted, and the simulation model of the DEH system was very close to the actual working situation of the system. Compared with the existing method of establishing a model, the method selected in this study is simpler and easy to achieve. The displacement of the oil engine, the oil pressure of the oil engine chamber, the oil supply pressure, and the oil discharge of the oil pump all conformed to the operation law of the steam turbine, which indicates that the model meets the requirements of simulation and research.

(2) A model-based method was used to identify the card astringency fault, and the data-based method was used to diagnose the leakage fault. The fault data collection and analysis show that the model can effectively identify the faults of the DEH system. The fault diagnostic model can almost simulate all the types of faults that occur in actual work, and it can diagnose unknown failure types.

(3) A multi-parameter gray error neural network prediction model was constructed by combining the advantages of the neural network and the gray system. The predicted value obtained from the gray system prediction model was taken as the input of the neural network, and the error between the expected value and the actual value was taken as the neural network's output. In the next work, we will further study the actual effect of the DEH system failure prediction of the turbine DEH system. Use the prediction model to determine the time it can know in advance before the fault.

[Comment #12]

The analysis and presentation of fault diagnosis and prediction section is not rigorous and not clear and it needs to rewritten in more clear way.

[Response #12]

The analysis in the Fault Diagnosis Prediction section has been revised to make it look more rigorous and clearer, pages 23-28.

[Comment #13]

The article is mainly developed according to other research, which is spread alongside the manuscript. Where is the novelty?

[Response #13]

Thanks for bringing up these critical points. We propose for the first time a prediction model for the DEH control system of a steam turbine, which is based on our established fault detection model, and it adopts the advantages of the gray system prediction model and the neural network system prediction, which is a multi-parameter gray error neural network prediction.

[Comment #14]

The block diagram of controlled system has to be put to explain the mechanism of control. Figure (2) does no show the whole control scenario. Discussion for it is also recommended.

[Response #14]

Figure (2) shows the structure included in the turbine simulation system, there are oil supply system, servo valves, fast unloading valves, oil motors, VCC voltage control card, it is the simulation model that we build out for collecting all the data of the turbine, it is the complete control scheme.

[Comment #15]

The choice of design parameters is still a challenging problem. If not possible to perform one of the modern techniques in tuning these design parameters, I suggest referring to some of them. 

[Response #15]

Thanks for the constructive suggestions. We have reconsidered the design parameters and they are indispensable.

[Comment #16]

References [29] are not cited in the text.

[Response #16]

All references have been revised of this manuscript.

[Comment #17]

In the simulation results section, the proposed approaches should compare with other relevant techniques which cited in the literature reviews. The authors are asked to further provide comparisons to better show the advantages of their approaches.

[Response #17]

Thanks for sharing this interesting point of view with the authors. We have added a portion of the citation to the paper and compared it.

[Comment #18]

The discussion of simulation is not based on reasoning. 

[Response #18]

The simulation is based on actual collected data, and we try to avoid the inference part in our new discussion in the paper.

[Comment #19]

The deep discussion is required in simulated results.

[Response #19]

Thanks for these excellent suggestions. We have discussed the results of the simulations and revised the manuscript.

[Line 210-213, Page 15] Jam fault

This simulation result is consistent with the results appearing on the 300MW steam turbine system [25]. The excellent nature and characteristics of SVM are applied in many areas. Even identifying financial time sequence prediction models [26].

[25] Zhang W Q, Zhang Y Y, Analysis on reasons of jam of high-pressure main stop valve in 300MW steam turbine[J]. Huadian Technology, 2011, 07,10-11(79).

[26] Yu B, Zhu Q, Yao J, et al. Design, mathematical modeling and force control for electro-hydraulic servo system with pump-valve compound drive[J]. IEEE Access, 2020, 8: 171988-172005.

[Comment #20]

The conclusion is descriptive. It is void of quantitative and numerical improvement and comparison and the conclusion has to be extended with future work.

[Response #20]

We have added the next steps in this manuscript to the conclusion of the paper and have endeavored to show the contribution this study has made over time.

[Line 426-428, Page 29] Conclusions

In the next work, we will further study the actual effect of the DEH system failure prediction of the turbine DEH system. Use the prediction model to determine the time it can know in advance before the fault.

[Comment #21]

The study did not address the practical verification of simulated results. The work is still weak unless it is experimentally implemented.

[Response #21]

We truly appreciate the reviewer’s valuable comments. We will conduct these studies in the future research, and the main objective of this study is to provide a methodology for DEH system failure prediction.

---

## [Decision Letter · Decision Letter 1]

4 Sep 2023

PONE-D-23-15933R1Intelligent Diagnosis and Prediction of Turbine Digital Electro-Hydraulic Control System Faults: Design and ExperimentationPLOS ONE

Dear Dr. Zhong,

Thank you for submitting your manuscript to PLOS ONE. After careful consideration, we feel that it has merit but does not fully meet PLOS ONE’s publication criteria as it currently stands. Therefore, we invite you to submit a revised version of the manuscript that addresses the points raised during the review process. Particularly those related to the embedded implementation.

We look forward to receiving your revised manuscript.

Kind regards,

Carlos Alberto Cruz-Villar, Ph. D.

Academic Editor

PLOS ONE

Journal Requirements:

Reviewers' comments:

Reviewer's Responses to Questions

**Comments to the Author**

1. If the authors have adequately addressed your comments raised in a previous round of review and you feel that this manuscript is now acceptable for publication, you may indicate that here to bypass the “Comments to the Author” section, enter your conflict of interest statement in the “Confidential to Editor” section, and submit your "Accept" recommendation.

Reviewer #2: All comments have been addressed

Reviewer #3: (No Response)

2. Is the manuscript technically sound, and do the data support the conclusions?

Reviewer #2: Yes

Reviewer #3: Partly

3. Has the statistical analysis been performed appropriately and rigorously? 

Reviewer #2: Yes

Reviewer #3: N/A

4. Have the authors made all data underlying the findings in their manuscript fully available?

Reviewer #2: Yes

Reviewer #3: No

5. Is the manuscript presented in an intelligible fashion and written in standard English?

Reviewer #2: Yes

Reviewer #3: Yes

6. Review Comments to the Author

Reviewer #2: I would like to thank the authors for revising the paper. All my comments have been addressed no further comments

Reviewer #3: The authors use the TensorFlow framework to accelerate intelligent diagnosis and prediction computing by employing an available ASIC to compute trained models. I suggest giving more insights into the embedded platform operated (Arduino, SparkFun, Board STM32F746, Himax WE-I Board, or whatever) and exposing the performance reached by this embedded platform.

It is essential to provide comprehensive information about the combination of TensorFlow with hardware, including a block diagram that clearly demonstrates the embedded platform's specific intervention in the overall system. This information can be found in lines 397-403.

7. PLOS authors have the option to publish the peer review history of their article (what does this mean?). If published, this will include your full peer review and any attached files.

Reviewer #2: No

Reviewer #3: No

---

## [Author Response · Author response to Decision Letter 1]

17 Oct 2023

Thank you for your specialized questions. We provide comprehensive information about combining TensorFlow with details how the embedded platform is implemented and its specific role in the overall system. 

[Line 404-410, Page 28-29]

As shown in figure 14, We started with the Edge TPU, which is a smart chip with an arithmetic power of 4 TOPS. We configured the runtime environment needed for the Edge TPU. Next, we installed the Tensorflow framework, and other dependent packages. Next, the configuration files were modified and compiled. Once this was done, we put the system code on the Edge TPU and ran it to diagnose and predict system failures. This embedded platform performs faster and with less loss of accuracy than the previous control.

[Figure 14, Page 29.]

---

## [Editor Report · Decision Letter 2]

31 Oct 2023

Intelligent Diagnosis and Prediction of Turbine Digital Electro-Hydraulic Control System Faults: Design and Experimentation

PONE-D-23-15933R2

Dear Dr. Zhong,

We’re pleased to inform you that your manuscript has been judged scientifically suitable for publication and will be formally accepted for publication once it meets all outstanding technical requirements.

Kind regards,

Carlos Alberto Cruz-Villar, Ph. D.

Academic Editor

PLOS ONE
---

## [Editor Report · Acceptance letter]

5 Nov 2023

PONE-D-23-15933R2 

Intelligent Diagnosis and Prediction of Turbine Digital Electro-Hydraulic Control System Faults: Design and Experimentation 

Dear Dr. Zhong:

I'm pleased to inform you that your manuscript has been deemed suitable for publication in PLOS ONE. Congratulations! Your manuscript is now with our production department. 

Kind regards, 

on behalf of

Dr. Carlos Alberto Cruz-Villar 

Academic Editor

PLOS ONE